# Stathmins and Motor Neuron Diseases: Pathophysiology and Therapeutic Targets

**DOI:** 10.3390/biomedicines10030711

**Published:** 2022-03-19

**Authors:** Delia Gagliardi, Elisa Pagliari, Megi Meneri, Valentina Melzi, Federica Rizzo, Giacomo Pietro Comi, Stefania Corti, Michela Taiana, Monica Nizzardo

**Affiliations:** 1Dino Ferrari Centre, Department of Pathophysiology and Transplantation (DEPT), University of Milan, 20122 Milan, Italy; delia.gagliardi@unimi.it (D.G.); elisa.pagliari@gmail.com (E.P.); rizzofederica18@gmail.com (F.R.); giacomo.comi@unimi.it (G.P.C.); mm.taiana@gmail.com (M.T.); 2Neurology Unit, Fondazione IRCCS Ca’ Granda Ospedale Maggiore Policlinico, 20122 Milan, Italy; megimeneri@gmail.com (M.M.); valentina.melzi85@gmail.com (V.M.); monica.nizzardo1@gmail.com (M.N.); 3Neuromuscular and Rare Diseases Unit, Department of Neuroscience, Fondazione IRCCS Ca’ Granda Ospedale Maggiore Policlinico, 20122 Milan, Italy

**Keywords:** stathmin, motor neuron diseases, ALS, SMA, STMN2, STMN1, axonal defects, cytoskeleton, microtubules

## Abstract

Motor neuron diseases (MNDs) are a group of fatal, neurodegenerative disorders with different etiology, clinical course and presentation, caused by the loss of upper and lower motor neurons (MNs). MNs are highly specialized cells equipped with long, axonal processes; axonal defects are some of the main players underlying the pathogenesis of these disorders. Microtubules are key components of the neuronal cytoskeleton characterized by dynamic instability, switching between rapid polymerization and shrinkage. Proteins of the stathmin family affect microtubule dynamics regulating the assembly and the dismantling of tubulin. Stathmin-2 (STMN2) is one of the most abundantly expressed genes in MNs. Following axonal injury, STMN2 expression is upregulated, and the protein is transported toward the growth cones of regenerating axons. STMN2 has a critical role in axonal maintenance, and its dysregulation plays an important role in neurodegenerative processes. Stathmin-1 (STMN1) is a ubiquitous protein that is highly expressed during the development of the nervous system, and its phosphorylation controls microtubule dynamics. In the present review, we summarize what is currently known about the involvement of stathmin alterations in MNDs and the potential therapeutic effect of their modulation, with a specific focus on the most common forms of MND, amyotrophic lateral sclerosis (ALS) and spinal muscular atrophy (SMA).

## 1. Stathmins Are Relevant for Axonal Stability

Motor neurons (MNs) are highly specialized cells equipped with long, axonal processes. Proper cytoskeletal structure is fundamental for maintaining shape, axonal stability, anterograde and retrograde transport and inter-neuronal signaling. Microtubules are essential for axonal outgrowth and regeneration and in maintaining the integrity of axonal signal transduction and cellular transport systems [1]. Axonal defects are some of the main players of the pathogenesis of motor neuron disorders (MNDs), and understanding the biology underlying these processes may increase the comprehension and the development of therapeutic targets in these diseases.

Microtubules are characterized by dynamic instability: they undergo periods of polymerization, shrinkage and rest, depending on the continuous balance between assembly and disassembly which is largely mediated by microtubule-associated proteins such as stathmins. 

The stathmin family includes four different phosphoproteins, stathmins 1–4, each encoded by different genes and mostly or exclusively expressed in the nervous system. Stathmin family members have a similar overall protein structure: all of them possess a C-terminal ‘‘stathmin-like domain’’ (SLD) that binds or releases tubulin dimers in a phosphorylation-dependent fashion, participating in the control of microtubule dynamics [2]. Their role is well described and studied in the oncological field, since the phosphorylation and/or dephosphorylation of stathmin regulates the dynamic balance of the microtubule network and the cell cycle [3]. It is for this role in the intracellular “relay” of extracellular signals mediated by phosphorylation that they have earned their name, which derives from the Greek “stathmos” (relay) [4].

The *STMN1* gene, located on chromosome 1, encodes for a 17kDa protein called Stathmin-1 (STMN1) or Oncoprotein 18 (OP18) (Figure 1). It is a ubiquitous, cytosolic phosphoprotein well characterized as a microtubule-binding protein with an important role in cellular functions, including mitosis, motility, process formation and intracellular transport [3]. It is expressed, differently from the other stathmins, in a wide range of cell types, including neurons, muscle cells and lymphocytes, and it is located in the cytoplasm mainly associated with the cytoskeleton. It possesses a conserved coiled-coil SLD at the C-terminal which binds and release tubulin after phosphorylation, but, in contrast with the other family members, it lacks the N-terminal intracellular membrane target [2,5]. Both STMN1 and STMN2 expression is finely regulated during the embryonic brain development; they are highly expressed during development with a peak at birth age and a decrease in adulthood, even if they still remain at high levels [6]. The most common role of STMN1 is its involvement in microtubule stability, exerted through interactions with tubulin. As already mentioned, this interaction is mediated by regulated phosphorylation; the most common modification occurs in Ser63, which leads to chemical change in the protein secondary structure, reducing its binding to α/β-tubulin heterodimers and, thus, suppressing the STMN1 inhibition activity of microtubules’ polymerization [7].

The *STMN2* gene, located on chromosome 8, encoded for Stathmin-2 (STMN2) or SCG10 (superior cervical ganglia 10), is a 21 kDa phosphoprotein specifically expressed in the nervous system (Figure 1). *STMN2* mRNA is particularly enriched in MNs, being one of the most abundant transcripts in these cell types [8]. It is mainly localized in the cytosol but is also linked to the Golgi apparatus, attached to trafficked vesicle membranes along axons and in axonal growth cones, where it promotes microtubule dynamics [9,10]. In contrast to STMN1, STMN2 has, in addition to the interactive SLD, a targeting domain at the N-terminal which mediates its membrane anchoring and, thus, its specific subcellular localization [9]. For this reason, it exists both in a soluble and in a membrane-bound form. Regulation of microtubule dynamics conducted by STMN2 is, again, tuned by the phosphorylation status of the protein. As with STMN1, it has some common phosphorylation sites, at Ser62 and Ser73, able to reduce, when phosphorylated, the microtubule-destabilizing role of STMN2. JNK1 (c-Jun N-terminal kinase 1) is one of the phosphatases that mostly influences the status of the protein acting on these two residues, thus, regulating microtubule homeostasis [10]. 

After injury, STMN2 is upregulated and recruited to growth cones of regenerating axons [11] by interacting with kinesin f1B (Kif1B); therefore, it has been proposed as a neuronal regeneration marker. Moreover, when depleted, STMN2 accelerates neurodegeneration, suggesting a role as mediator of axonal maintenance; its loss of function induces a paralytic phenotype [12] in Drosophila [13], and its mutant expression destabilizes neuromuscular junctions (NMJs) due to MN retraction [14]. Additionally, it has been observed that reduction of STMN22 in induced pluripotent stem cells (iPSC)-derived human MNs significantly impairs motor axon outgrowth and suppresses axonal regeneration, resulting in increased MN vulnerability [8,15]. 

However, high levels of stathmin have also been demonstrated to exert detrimental effects: both STMN1 and STMN2 overexpression in NSC-34 MNs lead to microtubule defects and Golgi alterations [16]. Therefore, a fine modulation of stathmin levels is likely to be fundamental for MN maintenance, and any alterations of its homeostasis might result in MN degeneration.

## 2. Stathmins in Disorders Affecting MNs

### 2.1. Stathmin and Vulnerable MNs

Amyotrophic lateral sclerosis (ALS) and spinal muscular atrophy (SMA) are the most relevant forms of MND, affecting both upper and lower MNs and exclusively lower MNs, respectively. However, within the same condition, not all MN populations are equally involved. Several studies demonstrated that, in both patients and animal models, some MNs are lost very early, while others remain intact, even at later stages [17,18,19,20]. In SMA and in both familial and sporadic forms of ALS, normal functions of oculomotor neurons (OMNs), as well as the Onuf’s nucleus, which controls sphincter function, are maintained throughout the late stages of the disease [21,22,23]. This selective MN preservation suggests that differential vulnerability between MN groups is largely independent of the causes of the disease and that mechanisms of vulnerability and resistance could be shared across different MNDs [24]. Identifying the common mechanisms underlying the selective protection of specific groups of MNs provides important biological insights into MN development and can also lead to the identification of novel, therapeutic targets for MNDs. Several studies examined gene expression patterns of vulnerable and resistant MN groups and identified molecular differences that may account for their observed, differential vulnerability [23,25,26].

In a recent study conducted by our group, synaptotagmin-13 (SYT13), a synaptic protein involved in vesicular metabolism and trafficking, was identified in rat and murine OMNs as a candidate gene for resistance to MN loss [27]. RNA-seq analysis of MNs in end-stage ALS patient tissues demonstrated SYT13 enrichment in the remaining resilient neurons in both OMNs and spinal cord MNs compared to controls. Similarly, in order to identify potential mechanisms of MN loss vulnerability, Kline et al. compared the transcriptome of vulnerable and resistant MNs, innervating abdominal and cranial muscles, respectively, from wild-type mice, and integrated it with three other, independent microarray screens [28]. They identified six transcripts common to the four datasets, including STMN1. Overexpression of these genes into a Drosophila model of ALS rescued the hallmarks of the neurodegenerative phenotype, demonstrating that the differentially expressed genes can function in disease-relevant pathways [28].

Similarly, insulin-like growth factor 2 (IGF-2) was found to be maintained in OMNs in ALS, potentially playing a role in oculomotor resistance in this disease [29]. Its receptor, IGF-1R, was highly expressed in OMNs and on the extraocular muscle endplate and binds IGF-1, mediating the activation of MN survival pathways in ALS. 

Taken together, these findings suggest that candidate genes protecting vulnerable MNs from degeneration might be identified by investigating oculomotor-specific expression.

### 2.2. Stathmins in Spinal Muscular Atrophy

The microtubular defects present in SMA were extensively studied by Wen and colleagues [30], who identified STMN1 as a crucial, pathogenetic factor for SMA. They first demonstrated that, after survival motor neuron (SMN) knockdown (KD), both in NSC34 cells and in the spinal cord and sciatic nerve of SMA-like mouse models, STMN1 levels were upregulated. In addition, they showed that STMN1 upregulation was correlated with disease severity in different SMA-like mouse types; STMN1 was significantly higher in type I and type II SMA-like mice, but no significant differences were seen in type III SMA-like mice versus controls. This finding was not supported by an increased transcription but, instead, by an enhanced stability of STMN1 in the SMA condition. Additionally, SMN loss induced a reduction in the proportion of tubulin polymers–without a decrease of the total α-tubulin levels–along with an impaired microtubule re-growth, which was caused by *STMN1* downregulation. They also confirmed these results in type I SMA-like mice, also providing a link between STMN1 and microtubule dynamics in vivo. This data were supported by the increased axonal outgrowth and recovery of mitochondrial density in SMA MNs after *STMN1* KD, also suggesting a role in organelle transport. 

The involvement of STMN1 in SMA was further investigated by Wen et al. [31], who generated a heterozygous (with 50% reduction of STMN1 protein) and a homozygous *STMN1*-knock-out (*STMN1*-KO) mouse model in a type III SMA-like background and in a control. As previously shown, they observed a decrease in the microtubule number in the sciatic nerve of SMA mice, which was partially rescued in heterozygous *STMN1*-KO mice, but not in homozygous *STMN1*-KO mice. Although there were no differences in survival time, body weight was found to be reduced between postnatal day 9 to 14 in SMA *STMN1*-KO mice compared to SMA-like mice. Additionally, heterozygous *STMN1*-KO mice presented improved motor performance until postnatal day 13. This was not supported by a difference in the MN number in the spinal cord or in SMN production, indicating that the reduction of STMN1 does not protect MNs from death and does not influence SMN production. Otherwise, the heterozygous *STMN1*-KO mice showed an amelioration of the NMJ defect compared to *STMN1*-KO and SMA-like mice. Lastly, they investigated the role of STMN1 as an agonist for Toll-like receptor 3 [32] and observed a milder gliosis in spinal cord sections of the heterozygous *STMN1*-KO mice compared to *STMN1*-KO and SMA-like mice, suggesting a possible role of STMN1 in neuroinflammation. Taken together, these studies suggest that the reduction of STMN1, not its complete absence, is required for SMA phenotype amelioration due to its role on axonal pathogenesis and the microtubule network. 

The role of STMN2 in SMA is still under investigation. Our group performed RNA sequencing of SMA iPSC-derived MNs compared with healthy controls and identified SMA-specific molecular pathways, including differential gene expression and splicing alterations in transcripts related to cytoskeletal, axonal and synaptic functions [33]. In particular, we found a deregulation of *STMN2*, confirming data obtained by microarray expression analysis in our previous work [34].

A motif-enrichment analysis enabled the identification of a common motif, motif 7, within the 3′ UTR of deregulated genes, including *NRXN2,* encoding Neurexin-2α, a presynaptic, cell-adhesion protein involved in signal transmission at synapses, and *STMN2* [33]. The RNA-binding protein (RBP) SYNCRIP (synaptotagmin-binding, cytoplasmic, RNA-interacting protein) is involved in mRNA stabilization and transcript modulation by binding motif 7 in 3′ UTR and interacting with full-length SMN [34]. Since SYNCRIP and full-length SMN protein bind *NRXN2* 3′UTR through motif 7 and increase its expression, it is possible that the missing interaction between SMN and SYNCRIP in SMA may be responsible for the *NRX2* reduction, with consequences on disease phenotype [33]. Given the presence of motif 7 in 3′UTR in *STMN2* mRNA, we can speculate that *STMN2* is regulated by the same mechanism of *NRXN2,* and its missed stabilization could, at least in part, account for axonal defects in SMA.

STMN1 was reported to be decreased in vulnerable MNs [28], acting as a potential disease modifier in MNDs. Intracerebroventricular administration of *STMN1* through adeno-associated virus (AAV)-9 was performed in the intermediate SMA model at postnatal day 2 [35]. After treatment, survival time and weight gain were significantly increased, and the righting reflex was improved in comparison with untreated SMA mice. Otherwise, no differences were seen in grip strength, motor performance and balance analysis. In support of the link between STMN1 and NMJ, an amelioration of neuromuscular innervation was detected after treatment in highly vulnerable muscles. Moreover, an enhanced spinal MN number and morphometry were observed compared to untreated SMA mice, along with an improvement of the filamentous density of the βIII-tubulin in the spinal cord in both untreated SMA and control mice. These last findings strengthen the link between STMN1 and the cytoskeleton regulation. Finally, they showed that AAV9-mediated treatment restored the levels of both α-tubulin and its acetylated form, suggesting that STMN1 does not affect axonal stability by acting on post-translational modifications, but restores the overall α-tubulin level.

These results appear in contrast with the findings by the other groups and were, at least in part, attributed to the use of a different mouse model and different techniques. However, this study adds new perspectives regarding the use of *STMN1* as a candidate gene for a combinatorial therapy in SMA. 

Given the role of STMN2 in the axonal stabilization, another possible therapeutic approach is targeting the levels of the functional protein. Indeed, Tararuk et al. demonstrated that STMN2 is regulated by phosphorylation, mediated by c-Jun N-terminal kinase 1 (JNK1) [36]. In this scenario, Klim and colleagues proved that JNK1 inhibition mediated by SP600125 led to an increase of STMN2 levels and resulted in amelioration of axonal outgrowth [15]. However, whether dysregulation in STMN2 is relevant in SMA pathogenesis and whether a therapeutic correction of *STMN2* levels might be useful for ameliorating the clinical phenotype should be further investigated. 

### 2.3. Stathmins in Amyotrophic Lateral Sclerosis

Axonal degeneration is an early feature of ALS that occurs long before symptom onset and MN loss [13]. Recent advances in the field of axonal biology demonstrate that survival and death of axons and neuronal cell bodies are driven by independent, underlying mechanisms [37]. Indeed, therapies that are able to preserve MN bodies but that do not prevent axons from degeneration failed to significantly ameliorate the pathological phenotype and to increase survival in animal models of this disease [38]. If axonal degeneration is a possible cause of disease, then axonal protection is likely to be a fundamental target for ALS, and a successful therapeutic strategy must be focused on both cell body and axonal protection.

Transactive response DNA-binding protein 43 KDa (TDP-43) is a nuclear DNA/RNA-binding factor involved in RNA-processing activities, including RNA transcription, splicing and transport [39,40]. Cytoplasmic mislocalization and aggregation of TDP-43 is the neuropathological substrate of ALS. 

A growing amount of evidence suggested the role of nuclear TDP-43 in regulating splicing integrity by suppressing nonconserved or cryptic exons [18]. Inclusion of cryptic exons in RNA transcripts induces translation disruption and nonsense-mediated decay [18]. Recently, Melamed and Klim independently found that *STMN2* is one of the most abundant transcripts in iPSC-derived MNs, and its correct splicing and expression is regulated by TDP-43 [8,15]. The *STMN2* gene contains five constitutive exons, along with a proposed alternative exon between exons 4 and 5. In normal conditions, TDP-43 binds to GU-rich sequences in the first intron of *STMN2*, inhibiting the usage of an alternative or cryptic polyadenylation site and prompting the normal splicing of intron 1. 

*STMN2* is extremely sensitive to TDP-43 depletion, since its levels decrease both after pharmacological TDP-43 relocalization from the nucleus and in *TARDBP*-mutated MNs derived from iPSCs or trans-differentiated from patients’ fibroblasts [8,15]. Indeed, mutations in *TARDBP*, the gene encoding TDP-43, or its depletion impair TDP-43 binding to *STMN2* pre-mRNA and result in the inclusion of a new, alternative, spliced exon within intron 1, called exon 2a, which, in turn, causes the insertion of a premature polyadenylation site and a stop codon. As a result, accumulation of a truncated and non-functional mRNA of *STMN2* occurs. 

Over 40 mutations in TARDBP have been identified in ALS patients; moreover, aggregation and loss of nuclear TDP-43 are the pathological hallmarks of the vast majority of ALS cases [41], suggesting that TDP-43 dysfunction and the subsequent, altered processing of *STMN2* could be a hallmark of ALS pathology. Indeed, reduced *STMN2* expression associated with the presence of the non-functional, truncated STMN2 was reported both in in vitro 2D models depleted of TDP-43 and in post-mortem brain and spinal cord tissues of familial and sporadic ALS patients with underlying TDP-43 pathology, including ALS patients with *Chromosome 9 Open Reading Frame 72 (C9Orf72)* repeat expansions, the most common genetic cause of ALS [8]. Conversely, altered splicing processing of STMN2 was not found in *Superoxide dismutase 1* (*SOD*1)-ALS patients that did not exhibit TDP-43 accumulation and mislocalization [42].

Aberrant *STMN2* mRNA processing, driven by knockdown of TDP-43 in iPSC-derived MNs, leads to neurite outgrowth and axonal regeneration defects similar to neurons with STMN2 depletion. Both direct restoration of *STMN2* through lentiviral delivery of *STMN2* gene without the cryptic polyadenylation site in the corresponding RNA [8] and JNK1 inhibition [15] rescue axonal phenotypes in *TARDBP*-depleted neurons, suggesting that reduction in STMN2 is sufficient to impair axonal function in ALS. This finding has pointed out a new, potential therapeutic target in the MND scenario. 

However, stathmin modulation must be carefully tuned, considering their importance in the dynamics of microtubule regulation. Transgenic *SOD1*-ALS mice showed increased levels of STMN1 and STMN2 in the spinal cord, and overexpression of both STMN1 and STMN2 in NSC-34 MNs has been proved to induce microtubule defects leading to Golgi fragmentation [16]. KD of *STMN1* and *STMN2* completely restored mutant, *SOD1*-linked molecular and morphological Golgi alterations in vitro, suggesting that maintenance of stable levels of these cytoskeletal regulators is fundamental for MN biology. 

Basal *STMN2* expression is characterized by natural heterogeneity, as suggested by the variability of STMN2 levels observed in sporadic ALS patients that did not correlate with the level of phosphorylated TDP-43 [8]. Theunissen and colleagues [43] linked this variability to the presence of *STMN2* variants, some of which were associated with increased risk of developing sporadic ALS. Specific *STMN2* variants may potentially determine the natural variability in the basal levels of *STMN2* transcription and could explain gene-expression variations that are independent from cryptic exon expression. Indeed, reduction of *STMN2* mRNA levels was also observed in sporadic patient olfactory neurosphere-derived cells, independent of the presence of phosphorylated TDP-43 [43]. Moreover, reduced *STMN2* expression was more recently reported in the brains of patients with Parkinson’s disease [44] and frontotemporal dementia [42] and in patients without TDP-43 pathology and in the absence of truncated *STMN2*, suggesting that *STMN2* expression could be regulated by other mechanisms that are different from the TDP-43-related, cryptic exon pathway. Loss of STMN2, then, with or without TDP-43 pathology, may represent a final, common pathway of neurodegeneration in ALS and in other diseases. 

## 3. Conclusions

Axonal and neuronal cell body loss are two distinct features of neurodegeneration and are prompted by independent, pathological processes. Disruption in microtubule dynamics is one of the earlier mechanisms involved in MNDs and is responsible for axonal degeneration. Stathmin protein structure contains a stathmin-like homology C-terminal domain able to bind tubulin and participates in the control of microtubule dynamics, mediating the assembly and disassembly of the axonal cytoskeleton. In turn, splicing and transcript stability of stathmin-2, one of the best-described members of the stathmin family, is finely regulated by a nuclear RNA-binding protein, TDP-43. Mislocalization and cytoplasmic accumulation of TDP-43 is the main hallmark of ALS/frontotemporal dementia neuropathology and causes several downstream processes due to both the gain of toxic function and the loss of its nuclear function. Specifically, TDP-43 depletion prevents the suppression of cryptic exons in *STMN2* mRNA, resulting in the production of truncated transcripts and non-functional protein isoforms. So, dysregulation of *STMN2* could represent the player on which cell body loss and axonal degeneration converge in ALS. 

Emerging clinical trials in ALS have increased the need for the development of new specific tools aiding diagnosis, prognosis estimation and monitoring of pharmacological response, aside from neurofilaments [45]. The assessment of neurophysiological indexes, such as the transcranial magnetic stimulation (TMS)-induced cortical silent period (CSP) [46], has improved the identification and the characterization of patients with ALS. In this clinical context, measurement of levels of prematurely truncated *STMN2* transcripts in patient biological fluids, especially cerebrospinal fluid (CSF), may provide an indirect, in vivo biomarker of TDP-43 pathology.

In addition, STMN2 might represent a promising therapeutic approach in the field of neurodegenerative disorders with underlying TDP-43 pathology. However, evidence of microtubular defects linked to increased *STMN2* expression was shown also in *SOD1*-ALS, in vivo models, suggesting that dysregulation of STMN2 might be the common, final pathway of neurodegeneration in ALS, independent from TDP-43 pathology. 

Alterations in several pathways dysregulated in different forms of ALS seem to converge on STMN2, suggesting that it could be a promising therapeutic target. However, since there is evidence that *STMN2* overexpression in MNs could lead to Golgi fragmentation due to microtubular defects, a delicate balance of STMN2 levels is required for MN homeostasis. This should be taken into account in view of the development of new molecular therapeutic strategies in ALS.

So far, the literature addressing the role of stathmin in SMA is scant. Although its role as microtubule regulator is well known, there is contrasting evidence on its function and its expression levels during the disease. SMN present some similarities with TDP-43 in terms of their subcellular localization and functions and contribute to the regulation of gene transcription and splicing by binding motif-7 on 3′UTR. Given the presence of motif 7 in 3′UTR in *STMN2* mRNA, it would be interesting to investigate whether *STMN2* expression could be regulated by the same mechanism and whether its dysregulation could take part in the pathogenesis of SMA. 

There is a growing amount of evidence suggesting that stathmins play a crucial role in neuronal homeostasis. STMN2 is strongly related to MNDs, being critically involved in the neurodegenerative cascade. Moreover, it has the potential to be used in the clinical field, both as a diagnostic and prognostic biomarker and as a molecular, therapeutic strategy. Further in vitro and in vivo studies are needed to elucidate the role of the stathmin protein family in a broader spectrum of neurodegenerative disorders.

## Figures and Tables

**Figure 1 biomedicines-10-00711-f001:**
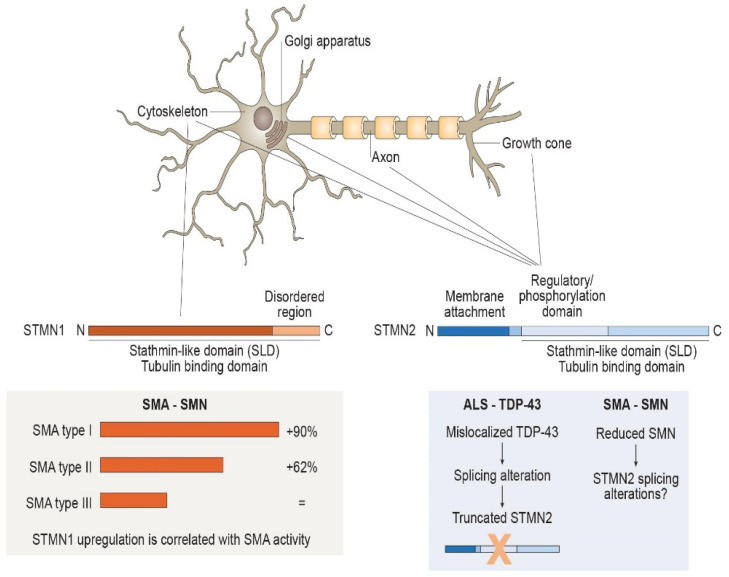
Structure, localization and regulation of Stathmin-1 (STMN1) and Stathmin-2 (STMN2). Stathmins regulate microtubule dynamics through phosphorylation-mediated binding of tubulin dimers with the C-terminal ‘‘stathmin-like domain’’ (SLD). STMN1 has a widespread cytosolic distribution, lacking the hydrophobic N-terminus. STMN1 protein levels are likely to be related to the presence of survival motor neurons (SMNs), since they are increased in spinal muscular atrophy (SMA) and seem to be correlated with disease severity. STMN2 presents hydrophobic residues at N-terminus that mediate its localization into intracellular membranes at the Golgi apparatus, associated with vesicles along the axons and within growth cones. STMN2 expression is regulated by transactive response DNA-binding protein 43 KDa (TDP-43), the mislocalization of which in amyotrophic lateral sclerosis (ALS) leads to STMN2-splicing alteration. STMN2 transcription and splicing could also be influenced by SMN considering its functional analogy with TDP-43 and, therefore, be involved in SMA pathogenesis.

## Data Availability

Not applicable.

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
