# Peer review of "Stathmins and Motor Neuron Diseases: Pathophysiology and Therapeutic Targets"

_biomedicines, 2022, doi:10.3390/biomedicines10030711_

Round 1
Reviewer 1 Report
In the paper “Stathmins and motor neuron diseases: pathophysiology and 2 therapeutic targets” the authors discuss the role of stathmin alterations in MDNs and possible interventions. To this aim the authors simplified existing models but preserved the fundamental anatomy. The manuscript reads timely and sound. The discussion offers a general overview of the findings, but it could be strengthened according to the following points.
To obtains a larger spectrum of “indirect in vivo biomarker of TDP-43 pathology”, in addition to biological measurements (STMN2), more systemic measures like the TMS-induced cortical silent period [a neurophysiological phenomenon widely used to assess the status of cortico-spinal motor neurons (Zeugin et al. 2021 Brain Sciences)], might complement the information brought by biological tests and contribute to broaden the overview of MND patients’ status.
It is suggested to better explain if/that the present study could be beneficial to illustrate how to ameliorate/boost rehabilitation, for example by complementing current intervention protocols (e.g. fostering a better focus on personalized interventions based on individual cortico-subcortico-spinal properties; Scandola et al 2019 – Journal of Neurotrauma) and/or improving clinical assessments (e.g. considering the variations in STMN2 as signs of the impact of neurological/psychiatric conditions).
Author Response
We thank the reviewer for his/her comments.
As suggested, we have integrated the manuscript by highlighting the relevance of neurophysiological measurements like TMS-induced cortical silent period in providing useful and “in vivo” knowledge about patients with MNDs.
We believe that stathmins could be markers of axonal damage in spinal/nerve injuries. However, the findings illustrated in this review are mostly derived from in vitro and in vivo models of MNDs and have not yet been fully translated to the clinical practice, particularly as tools to ameliorate rehabilitation
Reviewer 2 Report
Dear authors,
Congratulations, I enjoyed very much reading your work. I would only say that the figure should be a little higher/upper in the article. I mean before section 2. (Stathmins in disorders affecting MNs).
Best regards
Author Response
We thank the reviewer for his/her comments. We have moved figure 1 in the first section of the manuscript.
Reviewer 3 Report
The manuscript includes a detailed review regarding the role of Sathmin genes and proteins in motor neuron diseases (MNDs), in particular ALS and SMA, and the rational behind to propose them as therapeutic targets, based on the "to date" obtained research results.
The review is well organized and written.
I have some suggestions for authors that might improve the text. These are listed below:
- pg 2, line 80, there is an extra "g" letter, "..of g microtubule dynamics..."
- pg 4, line 178, the abbreviation NRXN2 has not been described, and the same for NRX2. The description of these genes/proteins could help the reader to better understand their role in the review's context.
- pg 4, line 191, there is an extra "T" letter.
- pg 5, line 225, "Transactive response DNA binding protein 43 KDa (TDP-43).
- pg 6, Figure 1, some abbreviations have been describes whereas other have not, e.g, TDP-43, ALS, SMN, SMA. Moreover, different color intensities in the red and blue lines might be explained.
- pg 7, line 306, FTD abbreviation has not been described in the text.
Author Response
We thank the reviewer for his/her comments. We modified the text accordingly.
Round 2
Reviewer 1 Report
Accept
Reviewer 3 Report
Authors have included all the suggestion in the new version of the manuscript.